# Development of a High-Quality/Yield Long-Read Sequencing-Adaptable DNA Extraction Method for Crop Seeds

**DOI:** 10.3390/plants12162971

**Published:** 2023-08-17

**Authors:** Naohiro Shioya, Eri Ogiso-Tanaka, Masanori Watanabe, Toyoaki Anai, Tomoki Hoshino

**Affiliations:** 1Laboratory of Crop Breeding, Graduate School of Agricultural Sciences, Yamagata University, 1-23 Wakaba-Machi, Tsuruoka 997-8555, Yamagata, Japan; a224171m@st.yamagata-u.ac.jp; 2Center for Molecular Biodiversity Research, National Museum of Nature and Science, 4-1-1 Amakubo, Tsukuba 305-0005, Ibaraki, Japan; 3Faculty of Agriculture, Yamagata University, 1-23 Wakaba-Machi, Tsuruoka 997-8555, Yamagata, Japan; mwata@tds1.tr.yamagata-u.ac.jp; 4Laboratory of Agroecology, Faculty of Agriculture, Kyushu University, 744 Motooka, Nishi-Ku, Fukuoka 819-0395, Fukuoka, Japan; anai.toyoaki.494@m.kyushu-u.ac.jp

**Keywords:** boom method, DNA extraction, long-read sequencing, seed, soybean

## Abstract

Genome sequencing is important for discovering critical genes in crops and improving crop breeding efficiency. Generally, fresh, young leaves are used for DNA extraction from plants. However, seeds, the storage form, are more efficient because they do not require cultivation and can be ground at room temperature. Yet, only a few DNA extraction kits or methods suitable for seeds have been developed to date. In this study, we introduced an improved (IMP) Boom method that is relatively low-cost, simple to operate, and yields high-quality DNA that can withstand long-read sequencing. The method successfully extracted approximately 8 µg of DNA per gram of seed weight from soybean seeds at an average concentration of 48.3 ng/µL, approximately 40-fold higher than that extracted from seeds using a common extraction method kit. The A_260/280_ and A_260/230_ values of the DNA were 1.90 and 2.43, respectively, which exceeded the respective quality thresholds of 1.8 and 2.0. The DNA also had a DNA integrity number value (indicating the degree of DNA degradation) of 8.1, higher than that obtained using the kit and cetyltrimethylammonium bromide methods. Furthermore, the DNA showed a read length N_50_ of 20.96 kbp and a maximum read length of 127.8 kbp upon long-read sequencing using the Oxford Nanopore sequencer, with both values being higher than those obtained using the other methods. DNA extracted from seeds using the IMP Boom method showed an increase in the percentage of the nuclear genome with a decrease in the relative ratio of chloroplast DNA. These results suggested that the proposed IMP Boom method can extract high-quality and high-concentration DNA that can be used for long-read sequencing, which cannot be achieved from plant seeds using other conventional DNA extraction methods. The IMP Boom method could also be adapted to crop seeds other than soybeans, such as pea, okra, maize, and sunflower. This improved method is expected to improve the efficiency of various crop-breeding operations, including seed variety determination, testing of genetically modified seeds, and marker-assisted selection.

## 1. Introduction

Crop breeding to improve crop yield is extremely important for supporting global population growth. In recent years, crop-breeding targets have been set not only for yield improvement but also for various other applications, such as achieving improved eating quality, high content of functional ingredients, and suitability for processing. Therefore, crop breeding is expected to become increasingly necessary to achieve breeding targets. Nucleotide sequence information, the blueprint of a crop, is extremely useful for crop breeding since it facilitates the creation of DNA markers and the extraction of genetic information, thereby accelerating the discovery of agriculturally useful genes that can then be used as breeding tools. Obtaining genomic DNA sequences from crops with high accuracy and at a low cost is becoming feasible owing to the rapid spread of next-generation sequencing (NGS) in recent years [1,2]. Short-read sequencing, which specializes in deciphering read lengths shorter than those that can be analyzed using the classical Sanger method [3], is now a mainstream NGS technique. However, using short-read sequencing, it is difficult to completely determine the location of highly repetitive sequences, which are often found in the large genomic DNA of crops with overlapping genomic DNA, such as soybeans. Even with targeted sequencing, which only sequences gene regions, it is difficult to analyze genes with similar sequences simultaneously. To solve this problem, long-read sequencing technology has recently been developed as a third-generation sequencer [4,5,6]. Long-read sequencing can generate sequenced DNA that is >100-fold longer than that obtained using conventional short-read sequencing technology. Unlike short-read analysis, which fragments DNA and uses a polymerase chain reaction to construct DNA libraries, thereby allowing sequencing from even small amounts of degraded DNA. On the other hand, long-read sequencing requires higher molecular weight DNA free of impurities at even higher concentrations than that required for conventional NGS analysis.

The DNA required for NGS analysis of crops is generally extracted from young leaves, and there are numerous kits and protocols for this purpose [7,8,9,10]. Recently, many quality DNA extraction methods suitable for long-read sequencing of leaves have been reported [11,12,13,14,15]. Young leaves offer the advantage of providing high-quality DNA, as they contain more DNA-rich young cells per unit weight. Additionally, DNA extraction from leaves requires seeding and cultivation, which can be disadvantageous. In contrast, crop seeds are considered unsuitable for DNA extraction because they are difficult to crush and contain many substances necessary for storage and dormancy, such as proteins, lipids, starch, polysaccharides, and secondary metabolites. However, DNA extraction from seeds for crop genomic research applications offers many advantages. For instance, it is unnecessary to grow a crop to obtain the leaves required for extraction. In addition, the seeds can be stored for long periods at low temperatures and used as samples whenever DNA extraction is required. Furthermore, because the seeds are stored, the DNA is not easily degraded when crushed at room temperature, eliminating the need to freeze the seeds and use liquid nitrogen. Despite these advantages, there are few optimal protocols or kits available for DNA extraction from crop seeds. Moreover, obtaining high-quality, high-molecular-weight DNA from seeds that can be used for long-read sequencing is expected to be even more difficult.

In this study, we present a method for extracting high-molecular-weight DNA from soybean seeds. We improved a scaled-up Boom method [16], in which soybean seed powder was dissolved in the presence of a chaotropic salt, guanidine thiocyanate, and DNA was adsorbed onto silica. This method does not require organic solvents such as chloroform or phenol, which are commonly used in the cetyltrimethylammonium bromide (CTAB) method, and is not subject to laboratory restrictions. The DNA extracted from the seeds had sufficient quality for Nanopore long-read sequencing, and similar results have been obtained from the leaves in previous studies [11,12,13]. We also showed that DNA can be extracted using this method from a variety of crop seeds other than soybean seeds in a simple manner, yielding high concentrations and purity.

## 2. Results and Discussion

### 2.1. DNA Extraction from Seeds Using the IMP Boom Method

In this study, we adapted the previously reported Boom method [16] for crop seeds, as summarized in Figure 1. To accelerate the lysis of crop cells, the amounts of surfactants in the lysis buffer were increased compared to those in the original Boom method. In addition, polyvinylpolypyrrolidone was added to the lysis buffer to remove impurities specific to crop cells. To improve DNA yield, the reaction system was scaled up to allow extraction from large sample volumes, and sample lysis time was significantly extended. Although the original Boom method concluded that size-fractionated silica particles were more efficient than diatomaceous earth for DNA extraction [16], we purposely used diatomaceous earth to improve the recovery of high-molecular-weight DNA. We modified diatomaceous earth washing into a two-step washing process to efficiently remove impurities while keeping the diatomaceous earth DNA adsorbed. The improved (IMP) Boom method could extract 3.29 μg/g seed weight of DNA at a concentration of 19.7 ng/μL from the soybean seeds when lysis was performed at 65 °C for 1 h, a step that dissolves cell and nuclear membranes (Table 1). DNA purity was evaluated in terms of the A_260/280_ and A_260/230_ values, which were much higher than the respective threshold values of 1.8 and 2.0 (Table 1). To improve DNA yield, we extended the lysis step and obtained 8.05 μg/g seed weight of DNA at a concentration of 48.3 ng/µL, as expected, while maintaining DNA purity (Table 1). Despite the expected increase in impurities due to the extended lysis step, the values indicating the purity of the DNA remained above the threshold values. This method for seeds extracted DNA with similar yields and high purity as that obtained from leaves (Table 1).

For comparison, soybean seeds and leaves were used to extract DNA using the kit and CTAB methods. Since many of the commercially available DNA extraction kits were prepared by modifying the original Boom method (16), one of those kits was used as a control for this study, as well as the CTAB method, a common method for DNA extraction from leaves. In the case of soybean leaves, the kit extracted 6.34 μg/g fresh weight of DNA at a concentration of 6.34 ng/μL, while the CTAB method extracted 75.7 μg/g fresh weight of DNA at a concentration of 454.0 ng/μL (Table 1). The A_260/280_ and A_260/230_ values of DNA obtained via the two methods were higher than the threshold values, except for the A_260/230_ value of the kit. The success of DNA extraction from leaves using the CTAB method is evidence that DNA extraction using leaves as samples is now an established technique.

On the other hand, when DNA was extracted from soybean seeds using the kit and CTAB methods, the extracted DNA was 1.25 μg/g seed weight at a concentration of 1.25 ng/μL and 24.2 μg/g seed weight at a concentration of 145.2 ng/μL, respectively (Table 1). The purity of the obtained DNA was poor, much below the threshold value. The CTAB method yielded higher yields than those obtained using the IMP Boom method, whereas the kit extracted very little DNA from the seeds. However, the seed-derived DNA solution extracted using the CTAB method was highly viscous (probably due to the high polysaccharide content) and difficult to dissolve, thus highlighting a need to determine whether it was structurally intact, of high quality, and free of foreign substances.

### 2.2. DNA Quality Analysis using Electrophoresis

To assess quality, the DNA extracted from soybean seeds and leaves using the kit, IMP Boom, and CTAB methods was electrophoresed on a 1.5% agarose gel. The DNA extracted using the IMP Boom and CTAB methods showed strong bands, indicating higher concentration, whereas the kit-extracted DNA showed weak bands (lower concentration) from both seeds and leaves (Figure 2A). The IMP Boom methods showed that DNA from both seeds and leaves was of high quality, with little smearing below the main band, degradation, and RNA. Despite RNase treatment in all the extraction methods, the CTAB method produced many smeared bands that appeared to be derived from RNA, especially in the seed-derived DNA solution, likely due to the incomplete degradation of RNA by RNase. The seed-derived DNA solution obtained using the CTAB method contained many impurities, and the CTAB method, which does not include DNA adsorption on silica, may have precipitated even small nucleic acids during centrifugation. Furthermore, while the CTAB method produced higher concentrations of both seed and leaf DNA, smeared bands were observed below the main bands, indicating that part of the DNA was degraded. Thus, we concluded that the DNA quality in the CTAB extraction was poor. After adjusting the DNA concentration of each sample to a constant level, the quality of the DNA was evaluated using a TapeStation 4200 system (Agilent Technologies, Santa Clara, CA, USA). DNA extracted using the kit and IMP Boom method showed only the main band, whereas DNA extracted using the CTAB method showed black smearing below the main band (Figure 2B). The DNA degradation index of the DNA extracted from seeds and leaves using the IMP Boom method was the highest upon a dissolution time of 3 h, and this value was higher than that obtained using the other two extraction methods (Table 1). These results indicated that the DNA extracted using the IMP Boom method is of high quality, with little DNA degradation occurring upon longer lysis times. To verify whether the Boom method, which has been IMP for extraction from seeds in this study, could stably extract DNA, DNA was extracted from 239 soybean seeds, and their yield and purity were determined. The DNA extracted had concentrations ranging from 12.1–75.1 ng/μL, with an average concentration of 41.1 ng/μL (Figure 3A). There were a few samples with extremely low DNA concentrations, which may have been due to DNA loss during the concentration and purification of DNA with isopropanol precipitation. On the other hand, the average values of A_260/280_ and A_260/230_ for these DNA samples were higher than the reference values, indicating high-quality DNA (Figure 3B). These results confirmed that the IMP Boom method could stably extract high-purity/concentration DNA from soybean seeds, although technical errors may occur when handling many samples.

### 2.3. DNA Quality Analysis using Long-Read Sequencing

To experimentally verify whether the DNA extracted from seeds using the IMP Boom method was truly highly pure and less degraded, long-read sequencing was performed using Oxford Nanopore. We carried out long-read sequencing as the success of this method and the read length of long-read sequences depend on DNA purity and molecular weight [11,12,13,14,15]. The DNA extracted via the IMP Boom and CTAB methods was sheared into approximately 30–35 kbp fragments using g-TUBE (Covaris, Woburn, MA, USA). Long-read sequencing was then performed (Figure 4A and Appendix A, Appendix A). During the library construction process, DNA extracted using the CTAB method resulted in the end-prep reaction mixture becoming cloudy, and the subsequent DNA yield significantly decreased after AMpure purification. During the AMpure purification after ligation, the library did not elute well in EB buffer, leading to a final yield of only about 1 ng/μL and a very low-concentration library (Appendix A). This suggested the presence of impurities in the seed-derived DNA solutions obtained using the CTAB method, which did not appear to be as pure as indicated by the absorbance values. Despite the low library concentration and significantly lower total number of reads and sequencing yield for CTAB-extracted DNA, we included it as a negative control since the read lengths were comparable to those in other analyses (Appendix A). The read length N_50_ of the sequence obtained using the IMP Boom method was approximately 14.7 kbp, whereas that obtained using the CTAB method was only about 1.0 kbp (Figure 4A). Furthermore, the IMP Boom method had a higher percentage of longer reads than the CTAB method, with a maximum read length of 75.1 kbp for the Boom method compared with 40.8 kbp for the CTAB method (Figure 4A). These results suggested that impurities in the DNA solution derived from seeds through the CTAB method may hinder the elution of high-molecular-weight DNA via AMpure. When a library of DNA extracted using the IMP Boom method was prepared without fragmentation and subjected to long-read sequencing, the read length N_50_ of the sequence increased to approximately 19.5 kbp, while the maximum read length increased to 127.8 kbp (Figure 4B). DNA extracted from seeds using the kit had a low yield, owing to which multiple samples were mixed for long-read sequencing; the read length N_50_ was 12.4 kbp, with a maximum read length of 73.2 kbp (Appendix A). On the other hand, DNA extracted using this kit showed the highest number of total reads, greatest sequencing yield, and highest mean and median read lengths for long-lead sequencing compared with DNA obtained via the IMP Boom and CTAB methods (Appendix A), although these results cannot be simply compared because they are considerably affected by differences in flow cell batches and reagents. Particularly, the Flongle flow cell is known for its unstable sequencing yield, requiring caution. While DNA extracted from seeds using this kit can be utilized for long-read sequencing, similarly to the IMP Boom method, it is not recommended for long-read sequencing due to the significantly lower yield of DNA extracted from seeds, which is crucial for nanopore sequencing that demands high amounts of DNA. Additionally, the low library concentration eluted after AMpure purification (Appendix A) suggested that impurities, as observed via the CTAB method, may also be present in the DNA solution obtained using this kit. These results indicated that the DNA extracted from seeds using the IMP Boom method is of high quality and can be used for long-read sequencing. In addition, this DNA was also adaptable for short-read sequencing in the whole-genome resequencing of soybeans (data not shown). Recently, several high-quality DNA extraction methods for long-read sequencing have been introduced [11,12,13,14,15], some of which have presented a read length N_50_ exceeding 50 kbp and a maximum read length exceeding 1.7 Mbp [14]. Although the present results are inferior to those reported in these studies, the DNA obtained from the seeds using the IMP Boom method is of sufficiently high purity, given that 10 kbp or more is defined as a long-read in NGS analysis. Many previous DNA extraction methods have been improved for plant leaf samples [11,12,13,14,15], but the same methods may not always be adaptable to seeds. Additionally, organic solvents are often used in the extraction process [12,14], and the loss of purity and degradation of DNA associated with this process are often high.

### 2.4. Comparison of Relative Amounts of Genomes in the Extracted DNA

To characterize the nuclear and organellar DNA in the extracted DNA, the relative amounts of each DNA were measured using quantitative real-time PCR (qPCR), with the DNA extracted using the kit, IMP Boom, and CTAB methods used as the templates. The ratio of mitochondrial DNA to nuclear genomic DNA was higher in the case of the CTAB method than in the case of the kit and IMP Boom methods for both the seed- and leaf-derived DNA solutions (Figure 5A). In contrast, the ratio of chloroplast DNA to nuclear genomic DNA was significantly lower and higher in DNA solutions extracted using the Boom and CTAB methods, respectively, as compared with that in the seed-derived DNA extracted using the kit, although there were no significant differences among the three methods for DNA solutions extracted from leaves. Interestingly, because the relative amounts of chloroplast DNA in the seed-derived DNA were significantly different among the three extraction methods, seed-derived DNA from the IMP Boom method had a significantly higher nuclear DNA content (Figure 5B). There is a need to determine the reason underlying the different DNA contents obtained using different extraction methods, only in the case of DNA derived from seeds. The silica used in the IMP Boom method is diatomaceous earth, which is expected to have a large surface area owing to its non-uniform shape, thereby possibly resulting in a higher affinity for polymeric DNA than that observed in the case of the kit. In contrast, the CTAB method does not use silica, which is expected to increase the relative amounts of small organellar DNA and RNA depending on the amount of nucleic acids and the presence of foreign material. When nuclear DNA is sequenced using NGS for both short- and long-read analyses, the organellar DNA present in the extracted DNA to be analyzed is an obstacle because it causes unnecessary sequence data to be acquired. These results indicated that the DNA extracted from seeds using the IMP Boom method reported in this study is not only pure enough to withstand long-read sequencing but also effective for NGS analysis because of the increased proportion of nuclear genomes.

### 2.5. Adaptation of the IMP Boom Method for DNA Extraction from Other Crop Seeds

To verify whether the IMP Boom method in this study can be adapted to other crop seeds, we used it to extract DNA from pea, okra, rice, maize, and sunflower seeds as well. The DNA extracted from all seeds, except rice, exceeded the yield of 8.33 μg/g seed weight at a concentration of 50 ng/μL, thus indicating high yield and good purity (Table 2). Agarose gel electrophoresis showed less smearing and degradation of DNA in the swim images, thus suggesting that this method can extract high-purity and high-yield DNA from many crop seeds with minimal DNA degradation (Figure 6). The low DNA yield obtained from the rice seeds could be attributed to the fact that most of the components of rice seeds are starch, and the membrane components are difficult to dissolve because of the viscosity of the lysis buffer solution. In this study, DNA was extracted from the seeds of other crops under conditions improved for soybean seeds. It is believed that high-purity and high-yield DNA extraction can be achieved by adjusting the sample volume at the time of lysis, the first step of the protocol, and the volume of the lysis buffer for each crop seed. Testing genetically modified crops requires DNA extraction from crop seeds to determine the presence of foreign DNA in the genomic DNA [17]. For the reason that only a small amount of DNA is required for PCR to test exogenous genes, a scaled-down adaptation of the IMP Boom method would lower costs and allow accurate DNA determination without false-negative results for genetically modified crops. In fact, we only attempted genotyping and amplifying gene regions using simple sequence repeat (SSR) markers on DNA from soybean seeds scaled down to 1/12th of the Boom method. Amplified PCR products of approximately 100–1000 bp of the target were adapted for all PCR experiments (Appendix A). These results indicated that low-cost DNA extraction using seeds could be used immediately for marker-assisted selection in crop breeding and gene function analysis. Our proposed IMP Boom method is not only adaptable to NGS analysis of long- and short-reads but also fully applicable to general-purpose laboratory experiments that are essential for genotyping and functional analysis of crop genes.

## 3. Materials and Methods

### 3.1. Plant Materials

Soybean (*Glycine max* (L.) Merr.) cultivar ‘Enrei’ was used as the experimental material for DNA extraction for long-read sequencing. Seeds were ground using a cool mill (TK-CM20S; Tokken, Chiba, Japan) at room temperature, then ground using a mortar and pestle at room temperature, and stored at 4 °C until DNA extraction. The leaves of 20-day-old soybeans were crushed using a mortar and pestle, frozen in liquid nitrogen, and stored at −30 °C until DNA extraction. The morning glory (*Ipomoea nil*), pea (*Pisum sativum*), okra (*Abelmoschus esculentus*), rice (*Oryza sativa*), maize (*Zea mays*), and sunflower (*Helianthus annuus*) seeds were ground in the same manner as the soybean seeds and stored at 4 °C until DNA extraction.

### 3.2. DNA Extraction

For the IMP Boom method, seed powder (about 0.3 g) was placed in a 15 mL centrifuge tube, and 6 mL of lysis buffer (containing 700 mM guanidine thiocyanate (Nakarai Tesque, Kyoto, Japan), 30 mM ethylenediaminetetraacetic acid (EDTA, pH 8.0), 30 mM Tris-HCl (pH 8.0), 0.5% Triton™ X-100 (Nakarai Tesque), 5% Tween-20 (Nakarai Tesque), and 1% polyvinylpolypyrrolidone (Sigma-Aldrich, St. Louis, MO, USA)) was added to it and mixed well with a vortex mixer. The lysed samples were incubated at 65 °C with stirring for 1 h. When the DNA yield improved, the incubation time of the samples was extended, as described below. The lysed samples were incubated at 65 °C for 3 h and then incubated overnight with stirring at room temperature. The incubated samples were centrifuged, and 5 mL of the obtained supernatant was collected in a 50 mL centrifuge tube. To this supernatant, 10 mL of binding buffer (5 M guanidine thiocyanate, 16.7 mM EDTA (pH 8.0), 8.3 mM Tris-HCl (pH 6.5), and 3.3% Triton™ X-100) with 25 mg silica (diatomaceous earth, Sigma-Aldrich) was added and stirred. After centrifugation, 5 mL of wash buffer 1 (3 M guanidine thiocyanate, 10 mM EDTA (pH 8.0), 5 mM Tris-HCl (pH 6.5), 2% Triton™ X-100, and 50% ethanol) was added after aspirating the supernatant while avoiding the precipitated silica. After another round of centrifugation, the supernatant was aspirated, and 15 mL of wash buffer 2 (50 mM NaCl, 10 mM Tris-HCl (pH 7.5), 0.5 mM EDTA (pH 8.0), and 60% ethanol) was added to it, and this procedure was repeated twice. Following centrifugation, the supernatant was aspirated and dried under reduced pressure. To the dried silica, 550 μL of 0.1× Tris-EDTA buffer containing RNase (Nippon Gene, Tokyo, Japan) was added and incubated at 65 °C for 5 min to elute the DNA. After centrifugation, the supernatant was transferred to a 1.5 mL tube.

To increase DNA concentration, 2-propanol precipitation was performed as described below. For 500 μL of DNA solution, 50 μL of 3 M sodium acetate and 500 μL of 2-propanol were added, stirred, and allowed to stand overnight at −30 °C. After centrifugation at 4 °C for 15 min, the supernatant was discarded, and 500 μL of 70% ethanol was added to it and stirred. This washing process was repeated twice. After centrifugation, the supernatant was discarded and dried under reduced pressure; the precipitate was dissolved in 0.1× Tris-EDTA buffer containing RNase.

For the CTAB method, seed powder (approximately 0.3 g) was placed in a 15 mL centrifuge tube, and 6 mL of 2% CTAB solution (100 mM Tris-HCl (pH 8.0), 20 mM EDTA (pH 8.0), 1.4 M NaCl, and 2% CTAB) was added and mixed well with a vortex mixer. The samples were then incubated at 65 °C while stirring for 3 h. 6 mL of chloroform was added to the incubated sample and stirred. After centrifugation, 5 mL of the supernatant was carefully collected in a 50 mL centrifuge tube, and 15 mL of 1% CTAB solution (50 mM Tris-HCl (pH 8.0), 10 mM EDTA (pH 8.0), and 1% CTAB) was added and gently stirred. After centrifugation, the supernatant was discarded, and 1 mL of 1 M NaCl containing RNase (Nippon Gene) was added to the precipitate to dissolve it, followed by incubation at 37 °C for 1 h. The incubated samples were concentrated and purified using 2-propanol precipitation, as described above.

For the DNA extraction kit, we used the Genomic DNA Extraction kit Min Plant (RBC Bioscience, New Taipei, Taiwan), which has shown a DNA extraction efficiency comparable to that of the widely used DNeasy Plant Maxi kit (Qiagen, Venlo, The Netherlands) and is less than half the price. DNA was extracted from 50 mg of leaf and seed powder samples, according to the manufacturer’s protocol.

### 3.3. Determination of DNA Quantity and Quality

DNA concentrations were quantified using a Qubit^®^ dsDNA HS Assay kit (Thermo Fisher Scientific, Waltham, MA, USA), according to the manufacturer’s instructions. The DNA quality was determined by measuring the absorbance at the wavelengths of 230, 260, and 280 nm using the NanoDrop™ One system (Thermo Fisher Scientific) for 1 μL of extracted DNA solution. To further examine the DNA quality using a different method, the extracted DNA solution was electrophoresed using a 4200 TapeStation (Agilent Technologies) to measure the DNA integrity number value, an indicator of DNA degradation. DNA quality was also checked by means of electrophoresis of 5 μL of extracted DNA solution on a 1.5% agarose gel with Tris/borate/EDTA and visualization after staining DNA with ethidium bromide (Nakarai Tesque).

### 3.4. Quantification of Nuclear, Chloroplast, and Mitochondrial DNA using qPCR

Nuclear, chloroplast, and mitochondrial DNA were quantified using qPCR based on previously reported methods [9,18]. qPCR was performed using the AriaMx Real-Time PCR System (Agilent Technologies) and Thunderbird™ SYBR^®^ qPCR Mix (Toyobo, Osaka, Japan). The relative amounts of nuclear, chloroplast, and mitochondrial DNA were determined using specific primer sets, as shown in Appendix A. PCR was performed at 95 °C for 1 min, followed by 40 cycles of 95 °C for 15 s and 60 °C for 1 min. Quantitative PCR products containing the target sequences of each primer set were used as copy-number standards. Genomic DNA content was quantified from copy ratio and genome size (nuclear, 987,495,272 bp; plastid, 152,218 bp; mitochondrial, 402,558 bp in Glycine_max_v4.0, available at https://www.ncbi.nlm.nih.gov/assembly/GCF_000004515.6/, accessed on 20 July 2022).

### 3.5. Long-Read Sequencing Analysis

DNA from Enrei was used for the long-read sequencing. It was fragmented using g-TUBEs (Covaris, Woburn, MA, USA), with two passages at 1300× *g* for 1 min in a centrifuge (TOMY, Tokyo, Japan). Raw and fragmented DNA (500 ng) were end-repaired and dA-tailed using the NEBNext^®^ End-Repair and dA-Tailing modules (New England Biolabs, Ipswich, MA, USA), according to the manufacturer’s instructions. The adapter was then ligated to the dA-tailed DNA using Quick T4 DNA Ligase (New England Biolabs), according to the manufacturer’s instructions, using SQK-LSK109, R9 version (Oxford Nanopore Technologies, Oxford, UK). Sequencing was performed using Flongle (R9.4) on a MinION™ Mk1c portable sequencer (both from Oxford Nanopore Technologies). The sequencing run was performed using MinKNOW software (version 22.12.7, Oxford Nanopore Technologies) with a minimum read length setting of 200 bp and the live-basecalling option disabled. The resulting FAST5 files in the “pass” folders, which correspond to sequences with high-quality scores, were converted to FASTQ files using Guppy (version 6.0.0+ab79250, Oxford Nanopore Technologies). Long-read sequencing was performed on two samples each for the seed-derived DNA extracted using the IMP Boom method with (Boom 1 and 2) and without (Boom 3 and 4) fragmentation, and the DNA extracted using the CTAB method (CTAB 1 and 2). Long-read sequencing was performed on one sample of DNA from seeds extracted using the kit. The resulting FASTQ files in the “pass” folders, which corresponded to sequences with high-quality scores, were used to calculate read length for Figure 4 and Appendix A.

### 3.6. Statistical Analysis

The significant differences in copy ratios of genome content and genomic content between the seed and leaf DNA extracted using the kit, IMP Boom, and CTAB methods were determined using Tukey’s HSD test (*p* < 0.05) with the multcomp package of R programming language version 4.1.2.

### 3.7. Genotyping with SSR Markers and PCR Amplification of the Soybean Gene Region

The SSR markers Satt181 and Satt373 were amplified using the DNA extracted from the seeds of the recombinant inbred line between Enrei and the edamame variety Shirayama as the template, using the IMP Boom method, with specific primers (Appendix A). PCR was performed using GeneAtlas^®^ G (Astec, Fukuoka, Japan) and GoTaq^®^ Green Master Mix (Promega, Madison, WI, USA), according to the manufacturer’s instructions. The thermal cycling conditions were 95 °C for 2 min; 35 cycles of 95 °C for 30 s, 60 °C for 30 s, and 72 °C for 30 s; and final extension at 72 °C for 3 min. To amplify the gene region in the DNA extracted from Enrei seeds using the IMP Boom method, the PCR products were amplified using specific primers for *GmActin* and *GmSACPD* (Appendix A), as previously reported [19]. PCR was performed using GeneAtlas^®^ G and PrimeSTAR^®^ GXL DNA Polymerase (Takara Bio, Shiga, Japan), according to the manufacturer’s instructions. The thermal cycling conditions were 98 °C for 3 min; 35 cycles of 98 °C for 10 s, 60 °C for 15 s, and 68 °C for 60 s; final extension at 68 °C for 3 min.

## 4. Conclusions

In this study, we propose an IMP Boom method that yields high-quality DNA suitable for long-read sequencing. The seed-derived DNA extracted using this method can be used in a variety of experiments required for crop breeding and genetic analysis. It should be noted that the IMP Boom method does not require the cultivation of crops to obtain extraction samples or use organic solvents. Moreover, the seeds can be crushed at room temperature, a unique advantage of seeds as a storage form over previous methods. For the reason that this method omits the cumbersome step of carefully collecting the supernatant, the actual working time was approximately 20 min, excluding the time required for cell lysis and drying the precipitate, which was approximately the same as the time required for DNA extraction using a typical kit. In addition, the cost per sample for this method was USD 2, which is comparable to or lower than the cost of the existing kit (Appendix A) and recently reported DNA extraction methods suitable for NGS analysis [7]. The IMP Boom method allows the simple, low-cost extraction of high-quality/yield DNA that is suitable for long-read sequencing. Such results are not achievable using other conventional DNA extraction methods for plant seeds. The IMP Boom method can also be adapted to the seeds of various crops other than soybeans; we believe that our results might further advance crop breeding research.

## Figures and Tables

**Figure 1 plants-12-02971-f001:**
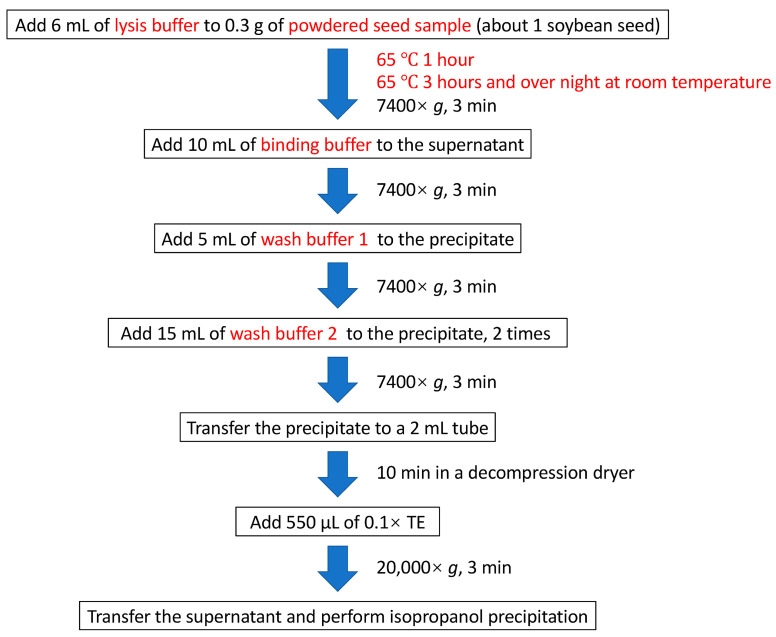
Schematic overview of DNA extraction carried out using the IMP Boom method. Details of the reagents used and the working process are described in Section 3. Processes improved from the original Boom method are shown in red.

**Figure 2 plants-12-02971-f002:**
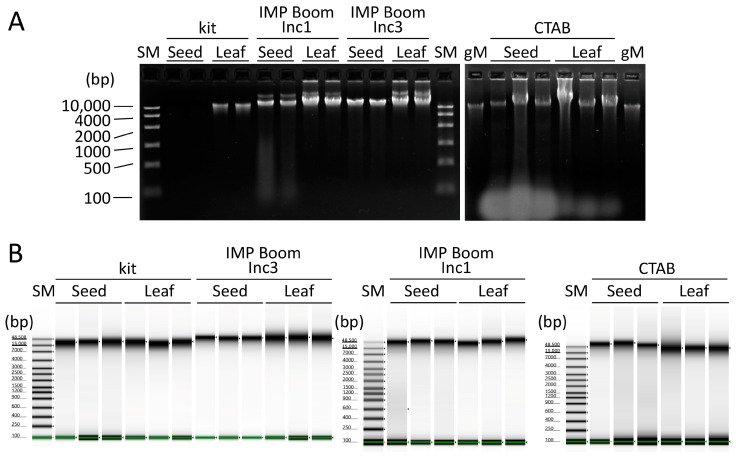
Quality assessment by means of electrophoresis of DNA extracted from soybean seeds and leaves using kit, IMP Boom, and CTAB methods. (**A**) Agarose gel electrophoresis using 1.5% agarose gel and 5 μL of extracted DNA solution. The Boom Inc1 row shows the DNA extracted through lysis at 65 °C for 1 h, while the Boom Inc3 row shows that extracted by means of lysis at 65 °C for 3 h, followed by overnight agitation at room temperature. SM is a size marker containing DNA lengths from 100–10,000 bp. gM is 10 ng/μL rice genomic DNA used as a comparison control. (**B**) Electrophoresis was carried out using TapeStation. SM is a size marker containing DNA lengths from 100–48,500 bp.

**Figure 3 plants-12-02971-f003:**
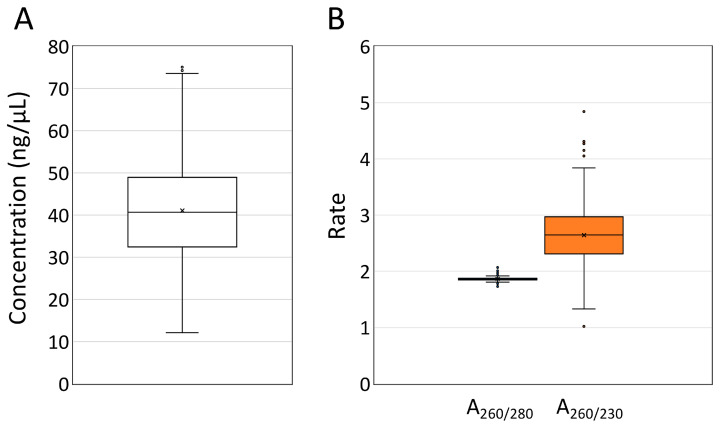
Evaluation of the yield and quality of DNA extracted from 239 samples of soybean seeds using the IMP Boom method. ’×’ in the figures indicate the mean values. (**A**) DNA concentrations estimated using Qubit^®^ (Thermo Fisher Scientific, Waltham, MA, USA). (**B**) A_260/280_ and A_260/230_ ratios calculated using Nanodrop^®^ One (Thermo Fisher Scientific).

**Figure 4 plants-12-02971-f004:**
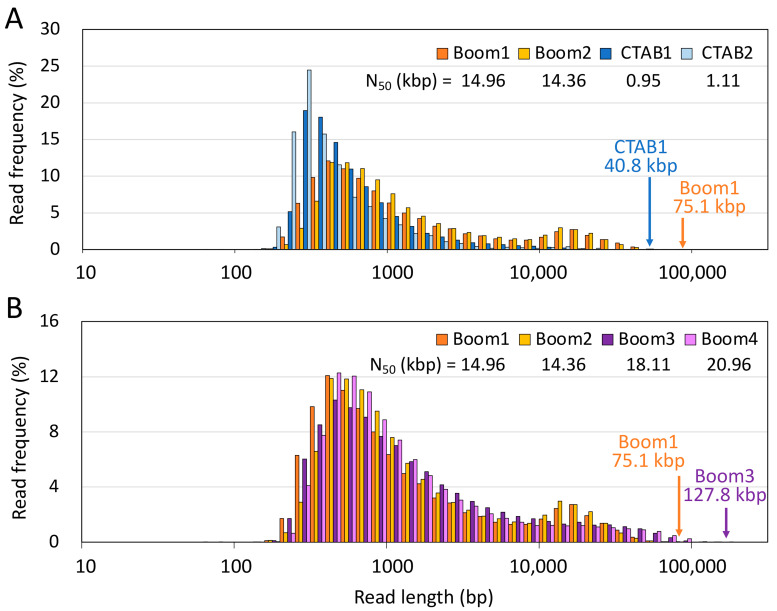
Read length frequency obtained using Oxford Nanopore sequencing. (**A**) Comparison of read length frequencies for DNA extracted using the IMP Boom and CTAB methods. Two long-read sequencing runs were performed on the DNA obtained using the two extraction methods. Boom1, 2 and CTAB1, 2 libraries were prepared by fragmenting DNA using g-TUBEs. The read length N_50_ values obtained from each library are shown in the upper-right corner of the figure. Orange and blue arrows indicate the maximum read lengths obtained using Boom1, 2 and CTAB1, 2, respectively. (**B**) Comparison of read length frequencies between library adjustments for DNA extracted using the IMP Boom method. Unlike the Boom1, 2 libraries, the Boom3, 4 libraries were prepared without fragmentation processing by g-TUBEs and long-read sequencing was performed twice. The read length N_50_ values from each library are shown in the upper-right corner of the figure. Orange and purple arrows indicate the maximum read lengths of Boom1, 2 and Boom3, 4, respectively.

**Figure 5 plants-12-02971-f005:**
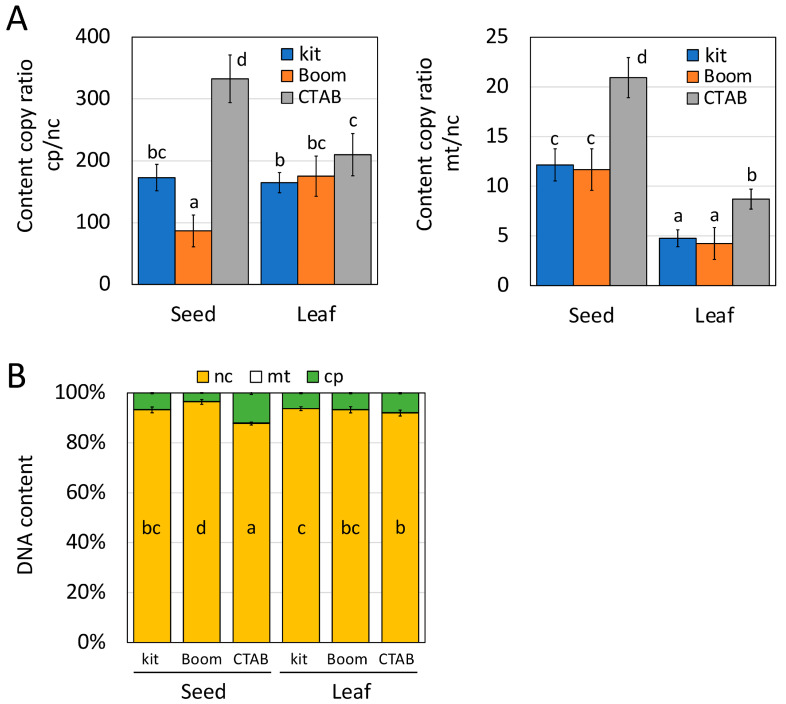
Comparison of genome copy-number for DNA extracted from soybean seeds and leaves using kit, IMP Boom, and CTAB methods. The different letters indicate statistically significant differences at *p* < 0.05, as determined using Tukey’s HSD test. (**A**) Chloroplast/nuclear genome and mitochondria/nuclear genome copy-number ratios. Blue, red, and gray indicate DNA extracted using the kit, IMP Boom, and CTAB methods, respectively. nc, mt, and cp indicate nuclear genomic DNA, mitochondrial DNA, and chloroplast DNA, respectively. (**B**) Percentage of genome composition in the extracted DNA. Orange, white, and green indicate nc (nuclear genomic DNA), mt (mitochondrial DNA), and cp (chloroplast DNA), respectively.

**Figure 6 plants-12-02971-f006:**
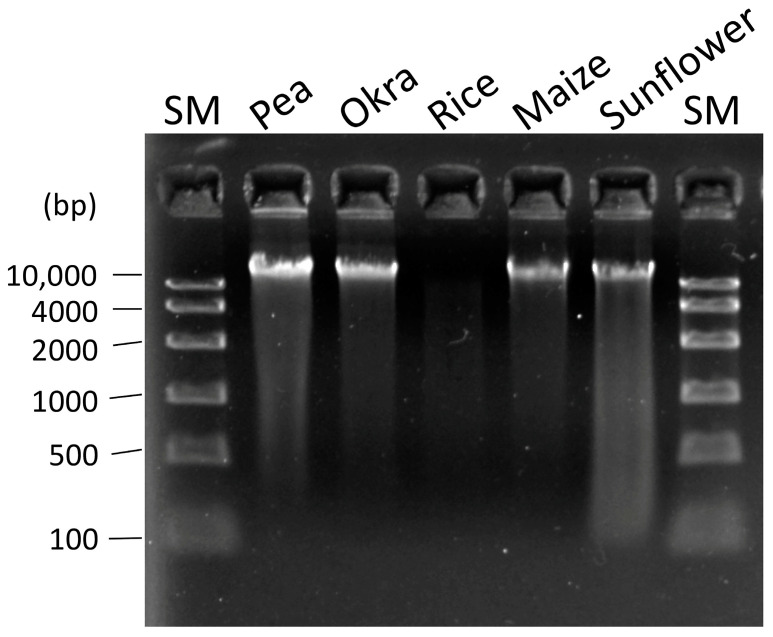
Agarose gel electrophoresis (1.5% agarose gel) of 5 μL DNA solution extracted from crop seeds using the IMP Boom method. SM is a size marker containing DNA lengths from 100–10,000 bp.

**Table 1 plants-12-02971-t001:** Yield and purity of the DNA extracted from soybean seeds and leaves using the kit, IMP Boom, and CTAB methods.

Method	Sample	Concentration (ng/μL)	Yield (μg/g)	A_260/280_	A_260/230_	DIN
kit	Seed	1.25 ± 0.12	1.25 ± 0.12	1.62 ± 0.05	1.44 ± 0.38	7.1 ± 0.17
	Leaf	6.34 ± 0.76	6.34 ± 0.76	1.81 ± 0.04	1.90 ± 0.04	7.5 ± 0.26
IMP Boom Inc1	Seed	19.7 ± 0.68	3.29 ± 0.11	2.10 ± 0.04	2.22 ± 0.06	6.9 ± 0.46
	Leaf	58.3 ± 4.54	9.71 ± 0.76	1.88 ± 0.02	2.26 ± 0.06	8.5 ± 0.75
IMP Boom Inc3	Seed	48.3 ± 5.15	8.05 ± 0.86	1.90 ± 0.02	2.43 ± 0.18	8.1 ± 0.23
	Leaf	48.9 ± 0.46	8.16 ± 0.07	1.93 ± 0.03	2.51 ± 0.08	8.8 ± 0.20
CTAB	Seed	145.2 ± 57.7	24.2 ± 9.6	1.90 ± 0.02	1.71 ± 0.09	7.0 ± 0.38
	Leaf	454.0 ± 71.9	75.7 ± 12.0	1.99 ± 0.02	2.66 ± 0.12	7.6 ± 0.40

Data are presented as mean ± SD of three biological replicates. The IMP Boom Inc1 row shows DNA extracted by means of lysis at 65 °C for 1 h, while the IMP Boom Inc3 row shows DNA extracted through lysis at 65 °C for 3 h followed by overnight agitation at room temperature. CTAB, cetyltrimethylammonium bromide. DIN, DNA integrity number value.

**Table 2 plants-12-02971-t002:** Yield and purity of DNA extracted from crop seeds using the IMP Boom method.

Sample	Concentration (ng/μL)	Yield (μg/g)	A_260/280_	A_260/230_
Pea	77.7 ± 8.48	12.95 ± 1.41	1.87 ± 0.02	3.63 ± 0.31
Okra	53.8 ± 14.81	8.97 ± 2.47	1.87 ± 0.01	3.54 ± 0.56
Rice	4.85 ± 0.49	0.81 ± 0.08	1.90 ± 0.02	2.31 ± 0.13
Maize	58.0 ± 6.15	9.67 ± 1.02	1.85 ± 0.01	5.94 ± 0.84
Sunflower	72.8 ± 10.86	12.13 ± 1.81	1.82 ± 0.07	5.42 ± 1.52

Data are presented as mean ± SD of three biological replicates.

## Data Availability

Data supporting the reported results are available on request from the corresponding author.

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
