# Peer review of "Development of a High-Quality/Yield Long-Read Sequencing-Adaptable DNA Extraction Method for Crop Seeds"

_plants, 2023, doi:10.3390/plants12162971_

Round 1

Reviewer 1 Report

The authors Shioya et al. introduced an approach, called optimized Boom method, which is relatively low-cost, simple to operate and yields high-quality DNA that can be used for long-read sequencing. The approach can extract 8ug of DNA per gram of seed weight from soybean seeds, at an average concentration of 48 ng/ul, which is approximately 40-fold higher than that extracted from seeds using a common extraction kit.

1. The term “optimized” refers to something that has been maximized/minimized as much as possible. As the proposed method is called “optimized Boom Method”, it would be expected that something has been maximized/minimized as much as possible.  The authors need to include - or justify why it is not needed - the original boom method in the comparison.

2.  It is actually surprising to see that a commercial kit would consistently generate negative A260/230 readings from 3 biological replicates of the leaf samples.  Was it due to Nanodrop error or contamination? 

3. The authors provided results from extraction of 239 soybean seeds in Figure 3.  The DNA concentration in fact showed a 6-fold variation (from 12.1 to 75.1 ng/ul).  Could the authors speculate what could be the reasons for such large variation?  Did all the seeds are of comparable sizes and are of the same soybean line?

4. The authors reported the read length frequency, N50 and the longest read. How about other parameters from Nanopore sequencing to assess the quality of the sequencing data, such as sequencing yield (total bases), number of reads, mean and median of read length? Would there be more results to support the quality of sequencing data from the optimized Boom method?

Author Response

The authors Shioya et al. introduced an approach, called optimized Boom method, which is relatively low-cost, simple to operate and yields high-quality DNA that can be used for long-read sequencing. The approach can extract 8ug of DNA per gram of seed weight from soybean seeds, at an average concentration of 48 ng/ul, which is approximately 40-fold higher than that extracted from seeds using a common extraction kit.

  1. The term “optimized” refers to something that has been maximized/minimized as much as possible. As the proposed method is called “optimized Boom Method”, it would be expected that something has been maximized/minimized as much as possible.  The authors need to include - or justify why it is not needed - the original boom method in the comparison.

Answer: We appreciate your comment regarding the use of the word "optimized." In response to your suggestion, we have stopped using the phrase "optimized Boom Method" in this paper because we did not implement the original boom method as a comparison. We have revised the title of this study to " Development of a High-Quality/Yield Long-Read Sequencing-Adaptable DNA Extraction Method for Crop Seeds". In this study, instead of including the original Boom method in the comparison, we used the Kit, which was modified from the original Boom method. Although the Kit cannot be compared strictly because its buffer composition, silica shape and degree of purification, etc. are not disclosed due to trade secrets, it is currently the method used by many researchers, so we considered it more suitable for comparison in this study than the original Boom method. This explanation of this point has been inserted in the text Line 120-123.

  1. It is actually surprising to see that a commercial kit would consistently generate negative A260/230 readings from 3 biological replicates of the leaf samples.  Was it due to Nanodrop error or contamination? 

Answer: We appreciate your comments on the data in the Table1. As you indicated, the negative values for A260/230 are not normal, so the negative values will be removed and marked as unmeasurable (Line 127). Since this measurement was performed consecutively with other samples on the same date and time, there cannot be the instrument error. The reason why only this sample has a negative value is unknown, but it is believed to be a contamination that exists only in this sample.

  1. The authors provided results from extraction of 239 soybean seeds in Figure 3.  The DNA concentration in fact showed a 6-fold variation (from 12.1 to 75.1 ng/ul).  Could the authors speculate what could be the reasons for such large variation?  Did all the seeds are of comparable sizes and are of the same soybean line?

Answer: Thank you for pointing about DNA concentrations. The soybeans used in this study are a recombinant inbred line of two varieties with different seed sizes, so the seed size varies from line to line, but we do not believe that the variation in DNA concentrations you noted is due to seed size. This method includes a step to concentrate and purify DNA by isopropanol precipitation, and we believe that some DNA was leaked during this step. We have inserted an explanation of this point in the text Line 178-184. Although such minor errors inevitably occur when experimenting with large numbers of samples, the average DNA concentration was 41.1 ng/μl after analyzing 239 samples, and we consider this method to be a stable extraction method.

  1. The authors reported the read length frequency, N50 and the longest read. How about other parameters from Nanopore sequencing to assess the quality of the sequencing data, such as sequencing yield (total bases), number of reads, mean and median of read length? Would there be more results to support the quality of sequencing data from the optimized Boom method?

Answer: We have data on sequencing yield (total bases) and the number of reads; however, we are unable to compare these values because the flow cell used in the study came from different batches. The sequencing yield and the number of reads is significantly influenced by the freshness and batch variability of the flow cell (especially Flongle flow cell), making it challenging to provide a meaningful evaluation. Therefore, we did not include these parameters in our result. However, based on empirical evidence, we found that the mean and median read length are minimally affected by flow cell batches. Hence, we have summarized these values in Table S1. Based on the mean and median read length results, the Kit was the best. However, since the Kit can extract very little DNA from seeds, the Kit cannot simply be rated highly based on these results alone. This explanation of this point has been inserted in the text Line 226-230.

Reviewer 2 Report

The manuscript ”Optimized Boom Method for Extraction of High-Quality/Yield Long-Read Sequencing-Adaptable DNA from Crop Seeds addresses an optimized Boom method for crop seeds, more specifically soybean seeds. The authors outlined a study that positively contributed to the optimization of DNA extraction methodology for this type of sample. In general, the theme proposed by the authors is of relativity importance. However, there is no novelty or innovation in this study. The study is well-designed and planned and the results and discussion are well presented and analyzed.

I recommend the publication of the manuscript with minor indicated amendments.

 Specific comments:

·         Abstract:

The abstract is well-written and gives the most important information provided by the study.

Lines 21 and 22 – correct the ratios.

Lines 33 – it's not novel, is an improved methodology.

 ·         Introduction: the study is clearly framed and presented in the topic under research.

Remove Lines 67 and 68 - However, leaf crushing requires liquid nitrogen, which must be prepared on a case-by-case basis.

 ·         Results and discussion

Lines 113 to 129 – this paragraph is a bit confusing. I understand it but sometimes to explain the differences between the kit and the CTAB method gets confusing.

Lines 139, 140, and 141 – 1.5% agarose - high agarose concentration- for genomic DNA generally 0.7%, maximum 1% is used; the bands are intense/soft or strong/weak not darker/lighter.

Line 224 – Additionally, organic solvents are often …

Line276– put ….nc (nuclear genomic DNA), mt mitochondrial DNA, and cp (chloroplast DNA) at the end of the caption of the figure5

 ·         Materials and methods:  the study was well planned, and the methodology was well described.

 ·         Conclusions: This study certainly provides valuable information for the optimization of DNA extraction methodology for this type of sample. However, the conclusions are a little ineffective and must be improved. the advantages should be mentioned again such as the applications, versatility, among others .... in another way to reinforce them.

Author Response

The manuscript ”Optimized Boom Method for Extraction of High-Quality/Yield Long-Read Sequencing-Adaptable DNA from Crop Seeds” addresses an optimized Boom method for crop seeds, more specifically soybean seeds. The authors outlined a study that positively contributed to the optimization of DNA extraction methodology for this type of sample. In general, the theme proposed by the authors is of relativity importance. However, there is no novelty or innovation in this study. The study is well-designed and planned and the results and discussion are well presented and analyzed.I recommend the publication of the manuscript with minor indicated amendments.

Answer: Thank you for your helpful remarks and comments. This paper is about the development of DNA extraction method that is an improvement of a previously published paper and may be inferior in novelty and innovation. However, DNA extraction is a basic and important operation in the field of crop breeding, including DNA marker selection, variety determination, and analysis of gene function. Furthermore, we believe that this extraction method will be useful to many readers because it can extract DNA that can be adapted to NGS analysis, which requires high quality, and can be adapted to other crop seeds besides soybean. We have made the following revisions to the comments received from the reviewer.

Specific comments:

  • Abstract:

The abstract is well-written and gives the most important information provided by the study.

Answer: Thank you for your positive comments.

Lines 21 and 22 – correct the ratios.

Answer: In response to your suggestion, we have noted the data in Table 1 accurately.

Lines 33 – it's not novel, is an improved methodology.

Answer: We have revised the description to say improved method, as you indicated.

・Introduction: the study is clearly framed and presented in the topic under research.

Answer: Thank you for your positive comments.

Remove Lines 67 and 68 - However, leaf crushing requires liquid nitrogen, which must be prepared on a case-by-case basis.

Answer: As you indicated, it has been removed.

・Results and discussion

Lines 113 to 129 – this paragraph is a bit confusing. I understand it but sometimes to explain the differences between the kit and the CTAB method gets confusing.

Answer: We have divided it into two paragraphs to avoid confusing the reader based on your point.

Lines 139, 140, and 141 – 1.5% agarose - high agarose concentration- for genomic DNA generally 0.7%, maximum 1% is used; the bands are intense/soft or strong/weak not darker/lighter.

Answer: The reviewer is correct. We also use 0.7% agarose gels when analyzing DNA fragments in detail on long gels, such as Southern hybridization. In this study, we used 1.5% gels, which we use regularly in our lab, because we performed electrophoresis to check the quality and concentration of high molecular weight DNA using short gels. In this case, 0.7-1.0% gels may be ideal, but even in 1.5% gels, the size marker (SM) can be checked up to 100-10,000 bp. We determined that the 1.5% gel was sufficient for checking DNA quality and concentration. We have also corrected the representation of bands on the gel as you suggested.

Line 224 – Additionally, organic solvents are often …

Answer: As you indicated, it has been corrected.

Line276– put ….nc (nuclear genomic DNA), mt mitochondrial DNA, and cp (chloroplast DNA) at the end of the caption of the figure5

Answer: As you indicated, it has been corrected.

・Materials and methods:  the study was well planned, and the methodology was well described.

Answer: Thank you for your positive comments.

・Conclusions: This study certainly provides valuable information for the optimization of DNA extraction methodology for this type of sample. However, the conclusions are a little ineffective and must be improved. the advantages should be mentioned again such as the applications, versatility, among others .... in another way to reinforce them.

Answer: As you pointed out, the conclusion mentioned the cost and simplicity of this method and did not describe the advantages of this study well. As you pointed out, we have corrected this error by again mentioning the strengths of this study.

Round 2

Reviewer 1 Report

1. The authors changed the wordings from "optimized Boom method" to "improved Boom method". But it is still modified from the Boom method, and the authors should include the information on what they have improved in the procedures or reagents of the extraction method. They used the commercial kit for comparison instead of the original Boom method. The reagents of the kit are not available for comparison due to trade secrets, yet the procedures could be compared, in which the kit uses silica membrane while their improved Boom method mainly uses precipitation and centrifugation. Could the authors at least comment on the yield and quality of extracted DNA based on the difference in procedures?

 2. For the negative value of A260/230, could the authors measure the values again to confirm that it is not due to Nanodrop error?  It is actually not an appropriate scientific practice to simply 'remove' results because they did not look good.  By removing a value from the table and mark it as unmeasurable, it makes all the other values in the table unreliable. If the authors believe that it is due to contamination, could they confirm whether it was due to the kit or the samples itself?

3. Lastly on the Nanopore sequencing data, those parameters including sequencing yield and total number of reads are important to show that Nanopore sequencing ran well.  It is a consensus that sequencing yields, number of reads, mean and median of read length are the basic parameters for measuring sequencing quality. It may not be comparable directly among different sequencing runs, but it is important to show them as supplementary result.

Overall, I think the authors were dodgy when addressing my comments.

Author Response

  1. The authors changed the wordings from "optimized Boom method" to "improved Boom method". But it is still modified from the Boom method, and the authors should include the information on what they have improved in the procedures or reagents of the extraction method. They used the commercial kit for comparison instead of the original Boom method. The reagents of the kit are not available for comparison due to trade secrets, yet the procedures could be compared, in which the kit uses silica membrane while their improved Boom method mainly uses precipitation and centrifugation. Could the authors at least comment on the yield and quality of extracted DNA based on the difference in procedures?

Answer: Thank you very much for your useful comments. Improvements from the original Boom method were described below. These explanations were inserted in the text Line 94-107.

1) To accelerate the lysis of crop cells, the amounts of surfactants (Triton X-100, Tween-20) in the lysis buffer were improved to increase from the original Boom method.

2) Polyvinylpolypyrrolidone (PVPP) was added to the lysis buffer to remove impurities specific to crop cells.

3) To improve DNA yield, the reaction system was scaled up to allow extraction from large sample volumes, and sample lysis time was significantly extended.

4) The original Boom method concluded that size fractionated silica particles were more efficient than diatomaceous earth for DNA extraction. We purposely used diatomaceous earth instead of silica particles to improve the recovery of high-molecular-weight DNA. As we expected, the IMP Boom method was able to extract high-molecular-weight DNA.

5) In the washing process of diatomaceous earth, we have modified it into a two-step washing process to efficiently remove impurities while keeping diatomaceous earth DNA adsorbed. In particular, the first washing process was improved by mixing ethanol with guanidine thiocyanate solution to prevent DNA loss.

While both of our IMP Boom methods and the Kit were similar in that they both improve on the original Boom method, the differences in the yield and quality of DNA extracted by the two methods were as follows.

1) The Kit can extract very little DNA from seeds, whereas the IMP Boom method yields high-quality/yield DNA. This explanation is found on line 136-138 of the manuscript.

2) Results of Nanopore analysis showed that DNA extracted from seeds using the IMP Boom method exceeded the values of Kit's DNA in read length N50 and maximum read lengths among the parameters of long-read analysis, and was determined to be of equal or better quality than Kit's DNA. This explanation is found on line 225-256 of the manuscript.

3) DNA extracted from seeds using the IMP Boom method showed an increase in the percentage of the nuclear genome compared to DNA extracted with Kit. The reason for this difference is not clear, but the silica used in the IMP Boom method is diatomaceous earth, which is expected to have a large surface area owing to its non-uniform shape, thereby possibly resulting in a higher affinity for polymeric DNA than that observed in the case of the Kit. This explanation is found on line 298-308 of the manuscript.

  1. For the negative value of A260/230, could the authors measure the values again to confirm that it is not due to Nanodrop error?  It is actually not an appropriate scientific practice to simply 'remove' results because they did not look good.  By removing a value from the table and mark it as unmeasurable, it makes all the other values in the table unreliable. If the authors believe that it is due to contamination, could they confirm whether it was due to the kit or the samples itself?

Answer: Thank you very much for your very informative comments. We have redone the measurements on the DNA extracted with the Kit using the Nanodrop again, as you suggested. The resulting A260/230 value in leaf DNA was positive value, as shown in the revised Table 1. Similarly, DNA extracted from rice seeds by the IMP Boom method was re-measured and found to be the positive value of A260/230, as shown in the revised Table 2. We have believed that the pre-revision results were not a machine error because they were measured on the same day in consecutive experiments. It is unclear why the result before revision was the negative value, but it may have been a measurement error due to mishandling of the elution buffer for zero value correction. We greatly appreciate your valuable comments leading us to re-analysis opportunity. These explanations were corrected on line 130-132 of the manuscript.

To verify that the re-measured values of leaf-derived DNA extracted from the Kit were indeed correct, we again extracted DNA from leaves using the Kit and measured yield and purity using Qubit and Nanodrop. This time, the amount of sample used was reduced by half, following the troubleshooting protocol; DNA concentration and yield were improved by reducing the sample volume by half. The A260/230 value for the DNA solution that was re-extracted was indeed positive value (Table R1). This result indicated that the Nanodrop used in this re-analysis worked properly and the values corrected in the manuscript were indeed correct. Because the re-extracted DNA solution was extracted at a different date and time than the DNA solution in the manuscript, this result is not included in the figures and tables in this manuscript and the supplement.

Table R1. Yield and purity of the DNA extracted from soybean leaves using the Kit by reducing the amount of leaf samples by half.

Method

Sample

Concentration (ng/μl)

Yield (μg)

A260/280

A260/230

Kit

Leaf

11.70±0.36

19.95±0.26

1.79±0.06

1.93±0.18

Data are presented as mean ± SD of three biological replicates.

  1. Lastly on the Nanopore sequencing data, those parameters including sequencing yield and total number of reads are important to show that Nanopore sequencing ran well.  It is a consensus that sequencing yields, number of reads, mean and median of read length are the basic parameters for measuring sequencing quality. It may not be comparable directly among different sequencing runs, but it is important to show them as supplementary result.

Answer: Thank you for your valuable comments. As you pointed out, we have presented all data including the sequencing yield, total number of reads in Table S1. We understand that these parameters are very important in demonstrating the success of Nanopore sequencing. In this study, we used different batches, different dates of the Flongle flow cells and reagents. We were concerned that presenting all the data would confuse our readers because there was such a large variation in the total number of reads from batch to batch, even though the length of the reads was relatively constant.

The primary objective of the long-read analysis in this study was to confirm whether the extracted DNA met the quality requirements for Nanopore sequencing, particularly in terms of obtaining high-molecular-weight DNA. For the DNA extracted using CTAB method, we observed a lack of sufficient library concentration, which suggested the presence of impurities hindering elusion from AMpure. Although the sequencing yield was very low due to the low library concentration, the read lengths were comparable to other analyses, which is why we included it as a negative control. On the other hand, the Kit showed the best parameters, except for read length N50 and maximum read length. However, the Flongle flow cell used for sequencing with the Kit appeared to have superior pore activity compared to those used for other analyses. While the total number of reads and sequencing yield cannot be directly compared because they are greatly affected by differences in flow cell batches and reagents, these results at least indicate that the DNA obtained by the IMP Boom method is of suitable quality for long-read analysis. Since the Kit can only extract a very small amount of DNA from the seeds, these results alone cannot be allowed for a simple high evaluation of the Kit. These explanations were inserted in the text Line 208-256.

Overall, I think the authors were dodgy when addressing my comments.

Answer: Thank you very much for your helpful comments to improve our manuscript. We have responded to your comments to the best of our ability.

Round 3

Reviewer 1 Report

They have properly addressed all my concerns. I am fine with this revision. They have properly addressed all my concerns. I am fine with this revision.